# Synthesis and Physicochemical Properties of 2,7-Disubstituted Phenanthro[2,1-*b*:7,8-*b’*]dithiophenes

**DOI:** 10.3390/molecules25173842

**Published:** 2020-08-24

**Authors:** Zhenfei Ji, Zeliang Cheng, Hiroki Mori, Yasushi Nishihara

**Affiliations:** 1Graduate School of Natural Science and Technology, Okayama University, 3-1-1 Tsushimanaka, Kita-ku, Okayama 700-8530, Japan; puf77a9l@s.okayama-u.ac.jp (Z.J.); pywp1wa6@s.okayama-u.ac.jp (Z.C.); 2Research Institute for Interdisciplinary Science, Okayama University, 3-1-1 Tsushimanaka, Kita-ku, Okayama 700-8530, Japan; h-mor@okayama-u.ac.jp

**Keywords:** phenacene-type compounds, thiophene ring, cross-coupling, alkyl side chains, UV-vis absorption, p-type organic semiconductors, organic field-effect transistor (OFET)

## Abstract

We report the design, synthesis, and physicochemical properties of an array of phenanthro[2,1-*b*:7,8-*b*’]dithiophene (PDT-2) derivatives by introducing five types of alkyl (C_n_H_2n+1_; n = 8, 10, 12, 13, and 14) or two types of decylthienyl groups at 2,7-positions of the PDT-2 core. Systematic investigation revealed that the alkyl length and the type of side chains have a great effect on the physicochemical properties. For alkylated PDT-2, the solubility was gradually decreased as the chain length was increased. For instance, C_8_-PDT-2 exhibited the highest solubility (5.0 g/L) in chloroform. Additionally, substitution with 5-decylthienyl groups showed poor solubility in both chloroform and toluene, whereas PDT-2 with 4-decylthienyl groups resulted in higher solubility. Furthermore, UV-vis absorption of PDT-2 derivatives substituted by decylthienyl groups showed a redshift, indicating the extension of their π-conjugation length. This work reveals that modification of the conjugated core by alkyl or decylthienyl side chains may be an efficient strategy by which to change the physicochemical properties, which might lead to the development of high-performance organic semiconductors.

## 1. Introduction

Since the development of the first organic field-effect transistor (OFET) in 1986, organic semiconductors have gained a great deal of attention because of their flexible, lightweight, and solution-process capable features [1]. Generally, by installing appropriate substituents, π-conjugated organic molecules with structural rigidity have shown significant properties, such as controllable solubility and light absorption ability, which are required for high-performance organic semiconductors [2,3,4,5,6]. Among them, the carrier mobilities of acene- and phenacene-based OFET reached over 30 cm^2^ V^−1^ s^−1^, typically with single-crystal devices [7,8,9]. Seeking to improve transistor properties, tremendous engineering progress has been made to modify the π-conjugated backbones in molecular design [10,11,12,13,14,15,16]. Side chains are usually introduced to affect the intermolecular packing and thin film morphology, leading to a suitable solubility, and thus, to high-performance devices [17,18,19,20,21,22,23,24,25]. Linear alkyl groups are the most commonly used as side chains in π-conjugated organic molecules [26]. A suitable installation of alkyl chains onto the conjugated backbones can increase the electronic coupling because of improved stacking in molecular aggregates [27]. In terms of the solubility of organic semiconductors, there are two factors; one is van der Waals interactions between side chains and the solvent, and the other is that the vibrational motions of the side chains result in a decrease in the intermolecular interactions between π-conjugated molecules [28,29,30]. Moreover, the significant effects of chain lengths, substitution positions, parity effect, and chirality on carrier mobilities have been investigated in different π-conjugated systems [31,32].

Recently, thiophene-containing fused molecules have played an important role in the progress of OFETs [33,34,35,36,37,38,39,40,41,42]. For instance, [1]benzothieno [3,2-*b*][1]benzothiophene (BTBT) [43], dinaphtho[2,3-*b*:2′,3′-*f*]thieno[3,2-*b*]thiophene (DNTT) [44], and their dialkyl derivatives formed a herringbone packing structure showing hole mobilities higher than 1 cm^2^ V^−1^ s^−1^. Moreover, the side chain effect of BTBT with alkyl groups of a different bulk on the charge transport properties was investigated, and devices based on BTBT bearing 2,7-di-*tert*-butyl groups exhibited high mobility above 10 cm^2^ V^−1^ s^−1^ [45]. Also, the incorporation of heterocycle linkages between the alkyl chains and conjugated backbones or heteroatom-containing side chains could significantly change the energy levels and molecular packing. The former is the introduction of alkylthienyl groups, which can increase the solubility and highest occupied molecular orbital (HOMO) energy levels. A solution-crystallized FET based on 2,9-bis(4-decylthiophen-2-yl)chryseno[2,1-*b*:8,7-*b’*]dithiophene (C_10_-Th-ChDT) exhibited hole mobility of up to 10 cm^2^ V^−1^ s^−1^ with reduced threshold voltage (*V*_th_) [46]. The latter is the use of alkylthio and alkylamino side chains, as reported by Zhang and coworkers, to enhance the hole and electron mobilities through the rise of HOMO and the improvement of sTable 2D molecular packing, which was partly the result of the overlap of p_π_(C)-d_π_(S) orbitals [47].

We previously reported the synthesis of phenanthro[1,2-*b*:8,7-*b’*]dithiophene (PDT) via Suzuki-Miyaura coupling of 3-formyl-2-thienylboronic acid with 1,4-dibromobenzene, followed by epoxidation and Lewis acid-catalyzed regioselective cycloaromatization (Figure 1, left) [48]. The introduction of two *n*-dodecyl groups into PDT (C_12_-PDT) along the longitudinal direction of the molecular axis showed a high crystallinity of its thin-film, resulting in a mobility as high as 2.2 cm^2^ V^−^^1^ s^−^^1^, i.e., higher by one order of magnitude than that of the parent PDT thin-film FET (1.1 × 10^−^^1^ cm^2^ V^−^^1^ s^−^^1^) [49]. This synthetic protocol can also be applied to the synthesis of the isomer of PDT, phenanthro[2,1-*b*:7,8-*b’*]dithiophene (PDT-2) (Figure 1, center). Likewise, the synthesized 2,7-didodecyl-substituted PDT-2 (C_12_-PDT-2) exhibited higher hole mobility, i.e., as high as 5.4 cm^2^ V^−1^ s^−1^, than that of C_12_-PDT, with a high-*k* gate dielectric [50]. This may be attributed to its favorable HOMO and HOMO−1 (hereafter NHOMO) geometries. To improve the hole mobility of PDT-2 derivatives, we reasoned that the introduction of a different array of alkyl or decylthienyl groups at 2,7-positions of the PDT-2 core may control crystallinity, solubility, and HOMO energy level, leading to improved transistor properties. We herein report the design, synthesis, and physicochemical properties of a series of 2,7-disubstituted PDT-2 derivatives (Figure 1, right).

## 2. Results and Discussion

From a viewpoint of the molecular design, the molecular orbitals for representative PDT-2 derivatives were calculated by the density functional theory (DFT) using Gaussian 09 package with a basis set of B3LYP/6-31G(d) (Appendix A) [51]. The HOMO and NHOMO (Figure 2), and HOMO and LUMO (Appendix A) of PDT-2, C_10_-PDT-2 and Th1-PDT-2 are shown with the dihedral angles (ψ) between the PDT-2 core and the decylthienyl groups. In our previous studies, molecular conformations showed a close relationship with HOMO and NHOMO distributions, while the length of the alkyl side chains had a negligible influence on HOMO and NHOMO coefficients. As seen in Figure 2, the introduction of decyl groups increased the HOMO level of C_10_-PDT-2 from −5.49 eV to −5.29 eV, which is the same tendency as that of C_12_-PDT-2 [50]. In contrast, the energy differences between HOMO and NHOMO and the large electron density localizing on sulfur atoms remained unchanged. This sulfur-dominated orbital is expected to make a great contribution to the electronic coupling [52]. Theoretical calculations of PDT-2 and C_10_-PDT-2 indicated that their HOMO and NHOMO coefficients are delocalized over the entire π-framework. For Th1-PDT-2 and Th2-PDT-2, the introduction of decylthienyl groups resulted in a big increase in the HOMO energy level, i.e., to −5.08 and −5.17 eV, respectively. This result might be due to the extension of π-conjugation length and the electron-donating effect of thienyl groups. Compared with Th1-PDT-2, Th2-PDT-2 showed a small increase in the HOMO energy level. The different sizes of these increases may be attributed to the degree of the dihedral angles between a PDT-2 core and a thienyl group. Th1-PDT-2 has a dihedral angle which is 10° smaller than that of Th2-PDT-2 (15°), resulting in a more efficient electron delocalization due to its extended π-conjugation, and a significant decrease in the electron density of the sulfur atoms of the PDT-2 core. The different dihedral angles may be caused by the different bond lengths of the terminal thiophene rings in the PDT-2 core and thienyl moieties between Th1-PDT-2 and Th2-PDT-2 (Appendix A). These slightly different bond lengths may cause steric repulsion between the PDT-2 core and decylthienyl groups.

The synthetic route of the target PDT-2 derivatives is shown in Scheme 1. We treated PDT-2 [50] with *n*-butyllithium, followed by the addition of bromine, resulting in 2,7-dibrominated PDT-2 **1** in 96% yield. Successively, five types of 2,7-dialkylated PDT-2 (C_n_-PDT-2: *n* = 8, 10, 12, 13, and 14) were synthesized in 42–67% yields by Suzuki-Miyaura coupling of **1** with various alkylboranes, derived from hydroboration of terminal alkenes and 9-BBN dimer. Additionally, using the Pd-catalyzed Migita-Kosugi-Stille coupling reaction, **1** was reacted with (5-decylthiophen-2-yl)- or (4-decylthiophen-2-yl)tributylstannane, resulting in the desired products, Th1-PDT-2 and Th2-PDT-2 in 52% and 61% yields, respectively.

To gain some insights into the effect of the side chains on the physical properties of the obtained five alkyl-substituted PDT-2 derivatives (C_n_-PDT-2), we measured their solubility in several organic solvents at room temperature. These compounds are poorly soluble in hexane and polar solvents such as acetonitrile and methanol. The correlation between the number of alkyl carbons and solubility of dialkylated PDT-2 derivatives in chloroform and toluene is shown in Figure 3. As predicted, in chloroform, the length of the side chains significantly affected the solubility, i.e., it gradually decreased with an increase in chain length [53]. C_8_-PDT-2 exhibited superior solubility in chloroform, i.e., 5.0 g/L, which is 10-fold higher than that of C_14_-PDT-2. This may be due to differences in the hydrophobic interactions of each molecule. Surprisingly, compared with the solubility of C_10_-PDT-2, the insertion of thienyl groups resulted in an approximately four-fold higher solubility of Th2-PDT-2 in toluene (Appendix A). In comparison, the substitution of 5-decylthienyl and 4-decylthienyl groups displayed significant changes in solubility that may be attributed to the difference of dihedral angles between the PDT-2 core and the thiophene ring, as shown in Figure 2. Th1-PDT-2 has a slightly smaller dihedral angle (ψ = 10°) that enhances the intermolecular π-orbital overlaps, resulting in poor solubility, whereas Th2-PDT-2 (ψ = 15°) exhibited higher solubility in both chloroform and toluene.

Next, the optical properties of PDT-2 derivatives were investigated by UV-vis absorption and fluorescence spectra in chloroform solution (Figure 4). Detailed data, including the absorption maximum wavelength (λ_max_^abs^), emission maximum wavelength (λ_max_^em^), absorption edge (λ_edge_), optical bandgaps (*E*_g_^opt^), and Stokes shifts, are summarized in Table 1. Figure 4a shows the absorption spectra of PDT-2 derivatives. The absorptions of dialkylated PDT-2 derivatives were nearly identical, with the strong peaks at 257, 276, 304, 324, and 339 nm, suggesting that the length of the alkyl chains has a negligible effect on the optical properties. All C_n_-PDT-2 showed very weak absorption at 365 nm, which corresponds to an S_0_ → S_1_ transition of picene-type molecules [23]. Since such a transition is forbidden, as is evident from TD-DFT calculations (Appendix A), their molar absorption coefficients (*ε*) are less than 1000 M^−1^ cm^−1^. In comparison, the parent PDT-2 shows a broad absorption band at about 272 nm, but not at 257 nm (Appendix A). Additionally, the introduction of alkyl chains resulted in a small redshift at 324 and 339 nm due to its electron-donating nature. For Th1-PDT-2 and Th2-PDT-2, the broad absorption bands were redshifted to 365 and 362 nm, respectively, along with the significantly increased absorption strength compared to those of dialkylated PDT-2 molecules. This result was due to the extension of the π-conjugation length of PDT-2. Their strong absorption bands, with a λ_max_ at about 380 nm, could be attributed to the π-π* transition, as predicted by TD-DFT calculations (Appendix A), whose lower energy absorption was attributed to the HOMO → LUMO transition. The longest absorption edge of C_n_-PDT-2 was located at about 370 nm, resulting in bandgaps of over 3.3 eV. Th2-PDT-2 exhibited the longest absorption edge of λ_edge_ at 404 nm, and the calculated *E*_g_^opt^ bandgap was 3.07 eV. The UV-vis absorption spectra indicated that thienyl groups can significantly lower the bandgap of the PDT-2 backbone. This side-chain effect can be explained by the electron-donating properties of thienyl groups. On the other hand, the results of the redshifted absorption edges and lower bandgaps were also attributable to the extended conjugated side chains attached to the PDT-2 backbone.

Furthermore, we investigated the UV-vis absorption spectra in the solid-state (Appendix A); the corresponding optical data are summarized in Table 1. The thin films of PDT-2 derivatives were prepared by spin-coating from hot chloroform solutions (ca. 0.5 wt%). In the case of dialkylated PDT-2 derivatives, they formed a heterogeneous, thin film. In contrast, the thin films of Th1-PDT-2 and Th2-PDT-2 were homogeneous and structureless. This indicated that Th1-PDT-2 and Th2-PDT-2 have better film-forming properties than those of the dialkylated PDT-2 derivatives, which are suitable for solution-processed OFETs. In sharp contrast, the absorption spectra of PDT-2 derivatives were broadened relative to the counterpart spectra in solution, and the vibrational peaks were red-shifted with respect to those in solution, suggesting the formation of intermolecular π-π stacking in the solid-state. In the thin films, the maximum absorption peaks of C_8_-, C_10_-, C_12_-, and C_13_-PDT-2 showed almost the same wavelength without significant differences in shape, indicating a negligible effect of the alkyl chain length (n = 8, 10, 12, and 13) on molecular packing. However, it is worth noting that C_14_-PDT-2 exhibited strong and obvious absorption peaks at 335, 352, and 371 nm, which differ from those of other alkylated derivatives with regard to the spectra shape, suggesting the formation of a well-ordered crystalline structure.

The fluorescence spectra of PDT-2 derivatives in chloroform are shown in Figure 4b. All of the dialkylated PDT-2 derivatives showed the same wavelength of emission maximum and similar emission peak shapes. The fluorescence spectra measured, using an excitation wavelength of 276 nm, exhibited strong intensity at 386 nm. Furthermore, C_n_-PDT-2 also exhibited an identical Stokes shift of 297 cm^−1^, whereas the fluorescence of Th1-PDT-2 and Th2-PDT-2 was characterized by an extended Stokes shift, i.e., more than three-fold that of others. The increasing Stokes shift was due to the introduction of flexible decylthienyl groups into the PDT-2 backbone, leading to reduced molecular rigidity and coplanarity. It is noteworthy that both Th1-PDT-2 and Th2-PDT-2 displayed significant fluorescence properties in chloroform, with Th1-PDT-2 emitting fluorescence bands at 399 and 422 nm, and Th2-PDT-2 emitting bands at 396 and 418 nm, which were excited at 365 and 362 nm, respectively.

## 3. Summary

In summary, various alkyl and decylthienyl-substituted PDT-2 derivatives were successfully synthesized and characterized. We found that alkyl chain length and types of side chains have a great effect on the physicochemical properties. For dialkylated PDT-2 molecules, the solubility was gradually decreased with an increase in carbon number, owing to increased hydrophobic interactions. The substitution with 5-decylthienyl groups exhibited poor solubility in both chloroform and toluene, whereas that with 4-decylthienyl groups resulted in higher solubility. All of these alkylated PDT-2 derivatives exhibited the proximate absorption maximum, suggesting that the change of alkyl chain length has a negligible influence on photophysical properties. The introduction of decylthienyl groups as conjugated side chains can slightly reduce bandgaps and increase HOMO energy levels. In the solid-state, all PDT-2 derivatives have broadened and red-shifted absorptions compared to the solution, indicating the formation of the ordered thin film. Among them, C_14_-PDT-2 exhibited the strongest and sharpest absorption peaks, suggesting the formation of a well-ordered crystalline structure. On the other hand, Th1-PDT-2 and Th2-PDT-2 had better film-forming properties than dialkylated PDT-2 derivatives, owing to their homogeneous and structureless nature, making them suitable for solution-processed OFETs. The PDT-2 derivatives presented in this work can thus be expected to serve as high-performance p-type semiconductors for OFET materials. Further evaluation of these derivatives for application as OFETs is currently in progress in our laboratory.

## 4. Experimental Sections

### 4.1. General

Unless otherwise noted, all reactions were carried out under an argon atmosphere using standard Schlenk techniques. Glassware was dried in an oven (150 °C) and heated under reduced pressure before use. Materials obtained from commercial suppliers were used without further purification. Solvents were employed as eluents for all other routine operations, and were purchased from commercial suppliers and employed without any further purification. For all thin-layer chromatography (TLC) analyses, Merck precoated TLC plates (silica gel 60 GF_254_, 0.25 mm) were used. Silica gel column chromatography was carried out using silica gel 60 N (spherical, neutral, 40–100 μm) from Kanto Chemicals Co., Inc. NMR spectra (^1^H and ^13^C{^1^H}) were recorded on Varian INOVA-600 (600 MHz). The chemical shifts were recorded in ppm relative to CDCl_3_ at 7.26 ppm and 1,1,2,2-tetrachloroethane-*d*_2_ at 6.00 ppm. The chemical shifts for ^13^C{^1^H} NMR were recorded in ppm downfield using the central peak of CDCl_3_ (77.16 ppm), 1,1,2,2,-tetrachloroethane-*d*_2_ (73.78 ppm) as the internal standard. Infrared spectra were recorded on a SHIMADZU IRPrestige-21 spectrophotometer and reported in wavenumbers (cm^−1^). UV-vis absorption spectra were measured using a Shimadzu UV-2450 UV-vis spectrometer. Fluorescence spectra were measured using SHIMADZU RF-5300PC. High-resolution mass spectrometry (HRMS) was carried out on a JEOL JMS-700 MStation (double-focusing mass spectrometer). Elemental analyses were carried out with a Perkin-Elmer 2400 CHN elemental analyzer at Okayama University. Geometry optimizations and normal-mode calculations were performed at the B3LYP/6-31G(d) level using the Gaussian 09, Revision D. 01, program package. PDT-2 was synthesized according to our previously reported procedure [50].

### 4.2. Synthesis of 2,7-dibrominated PDT-2 1

To a solution of PDT-2 (145 mg, 0.5 mmol), in anhydrous THF (15 mL) in a 20 mL Schlenk tube equipped with a magnetic stir bar under an argon atmosphere, was added dropwise n-butyllithium (1.6 M in hexane, 690 μL, 1.1 mmol) at −78 °C. After being stirred for 1 h at room temperature, the mixture was cooled to −78 °C again and bromine (62 μL, 1.2 mmol) was added dropwise. The reaction was stirred overnight at room temperature, quenched with water (5 mL), and poured into MeOH, which caused the precipitation of a pale yellow solid. The suspension was filtered, and the solid was dried under vacuum to yield **1** (210 mg, 96%). The spectroscopic and mass data were identical to those previously reported [50].

### 4.3. General Procedure for the Palladium-Catalyzed Suzuki-Miyaura Coupling of 1 with Alkylboranes

To a solution of 1-alkene (0.9 mmol), in anhydrous THF (6 mL) in a 20 mL Schlenk under argon, was added 9-BBN dimer (0.45 mmol) at room temperature. The reaction mixture was stirred at 60 °C for 1 h. Then, Pd(dba)_2_ (26 mg, 0.045 mmol), [HP^t-^Bu_3_]BF_4_ (26 mg, 0.09 mmol), powdered KOH (101 mg, 1.8 mmol), and **1** (134 mg, 0.3 mmol) were added successively at room temperature. The reaction mixture was stirred at 85 °C for 6 h, quenched with water (10 mL), and extracted with chloroform (30 mL × 3). The combined organic layers were washed with brine and dried over MgSO_4_. Filtration and evaporation yielded a brown solid. The residue was purified by column chromatography on silica gel (hexane:chloroform = 2:1), and subsequent recrystallization with acetone gave target dialkylated PDT-2 derivatives as a white solid.

*2,7-Dioctylphenanthro[2,1-b:7,8-b’]dithiophene (C_8_-PDT-2):* 55% yield. *R*_f_ = 0.79 (hexane:chloroform = 2:1). Mp = 263–264 °C. FT-IR (KBr, cm^−1^): 2956 (m), 2920 (s), 2873 (m), 2850 (s), 1465 (w), 1195 (w), 823 (w), 796 (s). ^1^H-NMR (600 MHz, CDCl_3_, rt): *δ* 0.88 (t, *J* = 7.2 Hz, 6H), 1.26–1.34 (m, 12H), 1.35–1.39 (m, 4H), 1.43–1.47 (m, 4H), 1.81–1.86 (m, 4H), 3.04 (t, *J* = 7.8 Hz, 4H), 7.74 (s, 2H), 8.00 (d, *J* = 9.0 Hz, 2H), 8.37 (s, 2H), 8.61 (d, *J* = 9.0 Hz, 2H); ^13^C{^1^H} NMR (150 MHz, CDCl_3_, rt): *δ* 14.3, 22.8, 29.36, 29.39, 29.5, 31.2, 31.7, 32.0, 119.0, 119.1, 121.1, 123.0, 126.6, 127.8, 136.9, 137.4, 147.4. Anal. Calcd for C_34_H_42_S_2_: C, 79.32; H, 8.22%. Found: C, 79.03; H, 8.24%.

*2,7-Didecylphenanthro[2,1-b:7,8-b’]dithiophene (C_10_-PDT-2):* 60% yield. *R*_f_ = 0.79 (hexane: chloroform = 2:1). Mp = 243–244 °C. FT-IR (KBr, cm^−1^): 2954 (m), 2918 (s), 2872 (m), 2846 (s), 1463 (m), 1192 (w), 835 (w), 821 (s), 792 (s). ^1^H-NMR (600 MHz, CDCl_3_, rt): δ 0.88 (t, *J* = 7.2 Hz, 6H), 1.26–1.32 (m, 20H), 1.35–1.38 (m, 4H), 1.43–1.47 (m, 4H), 1.81–1.86 (m, 4H), 3.04 (t, *J* = 7.2 Hz, 4H), 7.74 (s, 2H), 8.00 (d, *J* = 9.0 Hz, 2H), 8.37 (s, 2H), 8.61 (d, *J* = 9.0 Hz, 2H); ^13^C{^1^H} NMR (150 MHz, CDCl_3_, rt): *δ* 14.3, 22.8, 29.3, 29.5, 29.6, 29.7, 29.8, 31.2, 31.7, 32.1, 119.0, 119.1, 121.1, 123.0, 126.6, 127.7, 136.9, 137.4, 147.4. Anal. Calcd for C_38_H_50_S_2_: C, 79.94; H, 8.83%. Found: C, 79.82; H, 9.05%.

*2,7-Didodecylphenanthro[2,1-b:7,8-b’]dithiophene (C_12_-PDT-2)*: 67% yield. *R*_f_ = 0.79 (hexane:chloroform = 2:1). Mp = 233–234 °C. FT-IR (KBr, cm^−1^): 2954 (m), 2918 (s), 2870 (m), 2846 (s), 1463 (m), 1199 (w), 839 (w), 821 (s), 792 (s), 723 (w). ^1^H-NMR (600 MHz, 1,1,2,2-tetrachloroethane-*d*_2_, 80 °C): δ 0.92–0.95 (m, 6H), 1.32–1.39 (m, 28H), 1.44 (t, *J* = 7.2 Hz, 4H), 1.53 (m, 4H), 1.90 (m, 4H), 3.09 (t, *J* = 7.2 Hz, 4H), 7.78 (s, 2H), 8.04 (d, *J* = 8.4 Hz, 2H), 8.41 (s, 2H), 8.63 (d, *J* = 9.0 Hz, 2H); ^13^C{^1^H} NMR (150 MHz, 1,1,2,2,-tetrachloroethane-*d*_2_, 80 °C): *δ* 13.9, 22.4, 29.0, 29.1, 29.2, 29.3, 29.41, 29.42, 29.44, 30.9, 31.3, 31.7, 118.7, 118.7, 120.8, 122.8, 126.4, 127.5, 136.7, 137.2, 147.4. Anal. Calcd for C_42_H_58_S_2_: C, 80.45; H, 9.32%. Found: C, 80.63; H, 9.51% [50].

*2,7-Ditridecylphenanthro[2,1-b:7,8-b’]dithiophene (C_13_-PDT-2):* 42% yield. *R*_f_ = 0.79 (hexane:chloroform = 2:1). Mp = 224–225 °C. FT-IR (KBr, cm^−1^): 2954 (w), 2918 (s), 2872 (w), 2848 (s), 1463 (w), 1199 (w), 821 (m), 792 (m). ^1^H-NMR (600 MHz, 1,1,2,2-tetrachloroethane-*d*_2_, 80 °C): δ 0.92–0.95 (m, 6H), 1.32–1.39 (m, 32H), 1.44 (t, *J* = 7.2 Hz, 4H), 1.53 (m, 4H), 1.90 (m, 4H), 3.09 (t, *J* = 7.2 Hz, 4H), 7.78 (s, 2H), 8.04 (d, *J* = 8.4 Hz, 2H), 8.41 (s, 2H), 8.63 (d, *J* = 9.0 Hz, 2H); ^13^C{^1^H} NMR (150 MHz, 1,1,2,2,-tetrachloroethane-*d*_2_, 80 °C): *δ* 13.9, 22.4, 29.0, 29.1, 29.2, 29.3, 29.43 (2 carbons), 29.44, 29.5, 30.9, 31.3, 31.7, 118.7, 118.7, 120.8, 122.8, 126.4, 127.5, 136.7, 137.2, 147.4. Anal. Calcd for C_44_H_62_S_2_: C, 80.67; H, 9.54%. Found: C, 80.39; H, 9.33%.

*2,7-Ditetradecylphenanthro[2,1-b:7,8-b’]dithiophene (C_14_-PDT-2):* 62% yield. *R*_f_ = 0.79 (hexane:chloroform = 2:1). Mp = 216–217 °C. FT-IR (KBr, cm^−1^): 2954 (m), 2918 (s), 2870 (m), 2846 (s), 1462 (m), 1197 (w), 821 (m), 792 (m). ^1^H-NMR (600 MHz, 1,1,2,2-tetrachloroethane-*d*_2_, 80 °C): δ 0.95–0.97 (m, 6H), 1.35–1.41 (m, 36H), 1.43–1.47 (m, 4H), 1.52–1.56 (m, 4H), 1.90–1.93 (m, 4H), 3.10 (t, *J* = 7.2 Hz, 4H), 7.78 (s, 2H), 8.04 (d, *J* = 9.0 Hz, 2H), 8.40 (s, 2H), 8.62 (d, *J* = 9.0 Hz, 2H); ^13^C{^1^H} NMR (150 MHz, 1,1,2,2,-tetrachloroethane-*d*_2_, 80 °C): *δ* 13.9, 22.5, 29.06, 29.11, 29.2, 29.35, 29.44 (2 carbons), 29.46, 29.48, 29.49, 30.9, 31.3, 31.7, 118.7, 118.7, 120.8, 122.8, 126.4, 127.5, 136.7, 137.2, 147.3. Anal. Calcd for C_46_H_66_S_2_: C, 80.88; H, 9.74%. Found: C, 80.51; H, 9.60%.

### 4.4. General Procedure for the Palladium-Catalyzed Migita-Kosugi-Stille Coupling of 1 with (decylthiophene-2-yl)Tributylstannane

To a solution of **1** (140 mg, 0.31 mmol), in anhydrous DMF (7 mL) in a 20 mL Schlenk under argon, were added LiCl in THF (0.5 M, 1.56 mL, 0.78 mmol), (decylthiophene-2-yl)tributylstannane (0.78 mmol), and Pd(PPh_3_)_4_ (36 mg, 10 mol %). The reaction mixture was stirred at 100 °C for 10 h, then quenched with an aqueous solution of potassium fluoride at room temperature. The resulting suspension was extracted with chloroform (50 mL × 3). The combined organic layers were washed with brine and dried over MgSO_4_. Filtration and evaporation yielded a brown solid. The residue was purified by column chromatography on silica gel (hexane:chloroform = 10:1), and subsequent recrystallization with acetone gave the target product Th1-PDT-2 as a yellow solid.

*2,7-Bis(5-decylthiophene-2-yl)phenanthro[2,1-b:7,8-b’]dithiophene (Th1-PDT-2):* 52% yield. *R*_f_ = 0.60 (hexane:chloroform = 10:1). Mp = 239–240 °C. FT-IR (KBr, cm^−1^): 2954 (w), 2920 (s), 2850 (m), 1465 (w), 1190 (w), 819 (w), 792 (s). ^1^H-NMR (600 MHz, CDCl_3_, rt): *δ* 0.89 (t, *J* = 7.2 Hz, 6H), 1.29–1.43 (m, 28H), 1.73–1.75 (m, 4H), 2.82–2.90 (m, 4H), 6.77 (d, *J* = 1.2 Hz, 2H), 7.20 (d, *J* = 1.8 Hz, 2H), 8.01 (d, *J* = 9.0 Hz, 2H), 8.04 (s, 2H), 8.43 (s, 2H), 8.63 (d, *J* = 9.0 Hz, 2H); ^13^C{^1^H} NMR was not obtained due to its poor solubility. HR-MS (FAB^+^): Calcd for C_46_H_55_S_4_ [M + H] 735.3181. Found: 735.3185.

*2,7-Bis(4-decylthiophene-2-yl)phenanthro[2,1-b:7,8-b’]dithiophene (Th2-PDT-2):* 61% yield. *R*_f_ = 0.60 (hexane:chloroform = 10:1). Mp = 150–151 °C. FT-IR (KBr, cm^−1^): 2954 (w), 2916 (s), 2848 (s), 1460 (w), 1188 (w), 844 (w), 794 (s). ^1^H-NMR (600 MHz, CDCl_3_, rt): *δ* 0.89 (t, *J* = 7.2 Hz, 6H), 1.24–1.41 (m, 28H), 1.65–1.70 (m, 4H), 2.64 (t, *J* = 7.2 Hz, 4H), 6.92 (d, *J* = 1.2 Hz, 2H), 7.21 (d, *J* = 1.8 Hz, 2H), 7.98 (d, *J* = 9.0 Hz, 2H), 8.05 (s, 2H), 8.38 (s, 2H), 8.59 (d, *J* = 9.0 Hz, 2H); ^13^C{^1^H} NMR (150 MHz, CDCl_3_, rt): *δ* 14.3, 22.9, 29.51, 29.52, 29.6, 29.78, 29.80, 30.6, 30.7, 32.1, 117.9, 120.0, 120.3, 120.9, 123.2, 126.5, 127.0, 127.9, 136.9, 137.3, 137.7, 138.2, 144.5. HR-MS (FAB^+^): Calcd for C_46_H_55_S_4_ [M + H] 735.3181. Found: 735.3185.

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
