# Peer review of "Synthesis and Physicochemical Properties of 2,7-Disubstituted Phenanthro[2,1-*b*:7,8-*b’*]dithiophenes"

_molecules, 2020, doi:10.3390/molecules25173842_

Round 1
Reviewer 1 Report
Nishihara et al. reported the synthesis and physicochemical properties of PDT-2 derivatives. The alkyl chains on PDT-2 backbone has little impact on the electronic structures, but the solubility was decreased as the chain length was increased. On the other hand, decylthienyl-substituted PDT-2 derivatives show the red-shifted absorption. Interestingly, Th1-PDT-2 and Th2-PDT-2 exhibited clear difference in solubility. The results in this paper are interesting but several points should be addressed for the publication.
1) The dihedral angles for Th1-PDT-2 and Th2-PDT-2 affect their solubility and HOMO–LUMO gaps. The small dihedral angle provides the effective intermolecular π-π interaction and extension of π-system, which are consistent with the lower solubility and smaller Eg^opt for Th1-PDT-2. Why dihedral angles are dependent on the position of decyl group? The decyl groups does not seem to cause steric repulsions between PDT-2 and thienyl moieties.
2) Although the HOMO–LUMO transitions are forbidden for alkyl-substituted PDT-2 derivatives, the HOMO-LUMO transitions are allowed for decylthienyl-substituted PDT-2 derivatives. Why are the HOMO-LUMO transitions allowed for Th1-PDT-2 and Th2-PDT-2? Figure 2 shows that the orbital distributions of NHOMO for Th1-PDT-2 and Th2-PDT-2 are different from those for PDT-2 and C10-PDT-2. This suggests that the orbital distributions of LUMO can provide the explanation of allowed HOMO-LUMO transitions.
3) The fluorescence spectrum of C8-PDT-2 shows significantly larger intensity in 450-550 nm region compared to other alkyl-substituted PDT-2 derivatives. This implies that another excited species (e.g., aggregates or excimer?) can be generated for C8-PDT-2. The excitation spectrum (monitored at 500 nm) may provide further insights and discussion.
After the consideration and revision according to the comments by the reviewer, this paper may be accepted.
Minor points:
1) The optical properties of unsubstituted PDT-2 can be included in Table 1.
2) The chemical shifts of TCE-d2 should be provided in Experimental sections.
3) The reported 13C NMR data for C8-PDT-2, C14-PDT-2, and Th2-PDT-2 contain duplicated chemical shifts. When it is difficult to distinguish closely spaced peaks in 0.1 ppm precision, the chemical shifts should be reported in 0.01 ppm precision.
Author Response
Please see the attached pdf file.

Reviewer 2 Report
Review of the manuscript titled
Synthesis and Physicochemical Properties of 2,7‐3 Disubstituted
Phenanthro[2,1‐b:7,8‐ bʹ]dithiophenes
The article titled Synthesis and Physicochemical Properties of
2,7‐3 Disubstituted Phenanthro[2,1‐b:7,8‐bʹ]dithiophenes is connected
with synthesis and physicochemical properties of
phenanthrodithiophene derivatives. Conducted experiments revealed
that the alkyl length and the type of side chains have a significant effect
on their physicochemical properties. In the cause of dialkylated PDT‐2
molecules, the solubility was gradually decreased with an increase of a
carbon number, owing to increased hydrophobic interactions.
The substitution with 5‐decylthienyl groups exhibited poor
solubility in both chloroform and toluene, whereas that with 4‐
decylthienyl groups resulted in higher solubility. All of these alkylated
PDT‐2 derivatives showed the proximate absorption maximum,
suggesting that the change of alkyl chain length has a negligible
influence on photophysical properties. It is a crucial fact that designed
PDT‐2 derivatives presented in an article titled Synthesis and
Physicochemical Properties of 2,7‐3 Disubstituted Phenanthro[2,1‐
b:7,8‐bʹ]dithiophenes can thus be expected to serve as highperformance
p‐type semiconductors for the Organic Transistor
Materials (OFET).
The manuscript is connected with an exciting and modern
problem Organic Field Effect Transistor. The paper is written slightly
legible English. And from the scientific point of view, the text is
interesting enough. The literature review is careful and sufficient. The
graphic design - synthesis schemes, charts, spectra do not raise any
objections from the reviewer. Finally, the manuscript could be ready for
publication in the Molecules after minor revision.
Kind regards, Reviewer

Author Response
Please see the attached pdf file.

Reviewer 3 Report
The manuscript titled in “Synthesis and Physicochemical Properties of 2,7-Disubstituted Phenanthro[2,1-b:7,8-b’]dithophenes”, the authors reported the synthesis of various alkyl- and alkylthienyl-substituted PDT-2 derivatives and their basic properties, as a further work for these previous works (cited as ref [48]-[50]). Conjugated structures of the PDT-2 derivatives synthesized as semiconducting oligomers and the corresponding electrical properties should be included. However, the authors provided only basic characteristics such as solubility, UV-vis spectra of the derivatives in solution, and HOMO-LUMO bandgap. Since the conjugated structures of organic semiconductors significantly affect the electrical properties in the solid state, the authors should study film morphology and X-ray diffraction, as well as UV-vis spectra of these derivatives in a solid state. Based on the additional results, the authors need to discuss the originality of the new PDT-2 derivatives. A revised manuscript including all the supporting data will be considered if this work can be published as an Article in Molecules.
Author Response
We measured UV-vis absorption spectra of PDT-2 derivatives in thin-film, as shown in the new Figure S8, and the corresponding data are added in Table 1. Moreover, the description about the UV-vis absorption spectra in the solid state is also added in the revised manuscript. However, we could not measure the XRD and AFM measurements at this moment. We are very keen to report these measurements together with the FET-related properties of this paper when we report the XRD
and AFM results at other times.
